# Altered hippocampal interneuron activity precedes ictal onset

**Mitra L Miri[1][†], Martin Vinck[1][†], Rima Pant[1], Jessica A Cardin[1,2]\***

[1]Department of Neuroscience, Yale University School of Medicine, New Haven, United States; [2]Kavli Institute for Neuroscience, Yale University, New Haven, United States

**Abstract** Although failure of GABAergic inhibition is a commonly hypothesized mechanism underlying seizure disorders, the series of events that precipitate a rapid shift from healthy to ictal activity remain unclear. Furthermore, the diversity of inhibitory interneuron populations poses a challenge for understanding local circuit interactions during seizure initiation. Using a combined optogenetic and electrophysiological approach, we examined the activity of identified mouse hippocampal interneuron classes during chemoconvulsant seizure induction in vivo. Surprisingly, synaptic inhibition from parvalbumin- (PV) and somatostatin-expressing (SST) interneurons remained intact throughout the preictal period and early ictal phase. However, these two sources of inhibition exhibited cell-type-specific differences in their preictal firing patterns and sensitivity to input. Our findings suggest that the onset of ictal activity is not associated with loss of firing by these interneurons or a failure of synaptic inhibition but is instead linked with disruptions of the respective roles these interneurons play in the hippocampal circuit.
DOI: https://doi.org/10.7554/eLife.40750.001

**\*For correspondence:**
jess.cardin@yale.edu

[†]These authors contributed equally to this work

**Competing interests:** The authors declare that no competing interests exist.

## Introduction

Seizure activity is commonly considered to arise from an imbalance of excitation and inhibition in vulnerable neural circuits, leading to unconstrained activity that self-organizes into patterns of hypersynchrony. One mechanism of such an imbalance may be a transient loss of GABAergic inhibition (*Ziburkus et al., 2006*). Acute blockade of GABA receptors rapidly initiates seizure activity (*Rose and Blakemore, 1974*; *Treiman, 2001*), suggesting the necessity of synaptic inhibition to maintain healthy activity patterns. Previous work has also highlighted the potential preictal contribution of excitatory GABAergic effects due to Cl- accumulation (*Cossart et al., 2005*; *Palma et al., 2006*; *Miles et al., 2012*) or loss of inhibition due to depletion of GABAergic release (*Zhang et al., 2012*). Interneuron firing may also transiently cease during ictal events as a result of depolarization block (*Ziburkus et al., 2006*; *Cammarota et al., 2013*; *Kim and Nykamp, 2017*). Developmental dysregulation of inhibitory interneurons causes chronic epileptic disorders (*Lau et al., 2000*; *Cobos et al., 2005*; *Rossignol et al., 2013*; *Tai et al., 2014*), and interneurons also appear to be particularly sensitive to seizure-related damage (*Sloviter, 1987*; *de Lanerolle et al., 1989*; *Robbins et al., 1991*; *Rice et al., 1996*; *Gibbs et al., 1997*; *Cossart et al., 2001*). However, the circuit mechanisms underlying seizure initiation in vivo and the specific role of GABAergic interneurons remain largely unknown.

Work from in vitro and in vivo animal models has suggested that different neural populations may have distinct and complex patterns of preictal activity during seizure initiation. Putative interneurons in multiple hippocampal regions are reported to exhibit increases and/or decreases in firing rates prior to seizure initiation (*Grasse et al., 2013*; *Toyoda et al., 2015*). In vitro recordings during seizure-like events further suggest interleaved bursts of firing by interneurons and excitatory neurons (*Ziburkus et al., 2006*). In the cortex, fast-spiking, putative inhibitory interneurons may exhibit

enhanced preictal firing (*Timofeev et al., 2002*; *Gnatkovsky et al., 2008*), but recent work using population imaging largely observed increased activity following seizure onset (*Khoshkhoo et al., 2017*). Putative fast-spiking interneurons in human cortex may also exhibit increased preictal firing rates (*Truccolo et al., 2011*). In comparison, previous work has largely found increased preictal firing of putative excitatory hippocampal neurons (*Cymerblit-Sabba and Schiller, 2010*; *Jiruska et al., 2010*; *Fujita et al., 2014*), but several reports have highlighted mixed populations, with some neurons increasing and some decreasing firing prior to seizure (*Bower and Buckmaster, 2008*; *Toyoda et al., 2015*). The trajectory of interneuron activity and the relationship between excitatory and inhibitory neurons during seizure initiation thus remain unclear.

One key challenge in examining the respective roles of excitatory and inhibitory cells in seizure initiation is the diversity of inhibitory interneurons. Neocortical GABAergic interneurons exhibit a wide variety of morphologies, molecular markers, and activity patterns, and make synapses on different subcellular domains of target pyramidal cells (*Rudy et al., 2011*). In the hippocampus, these include the soma-targeting, axo-axonic and basket cells that co-express the calcium binding protein parvalbumin (PV) and the dendrite-targeting O-LM and bistratified interneurons that co-express the peptide somatostatin (SST) (*Buhl et al., 1994*; *Sik et al., 1995*; *Klausberger et al., 2003*; *Petilla Interneuron Nomenclature Group et al., 2008*; *Lapray et al., 2012*). Despite their divergent cellular properties, these cell classes are difficult to identify in vivo using traditional recording methods. Conflicting previous reports of the preictal activity of putative interneurons may thus be due in part to recordings of mixed neural populations.

Here, we used optogenetic tools to identify, track, and probe two distinct populations of hippocampal interneurons, the PV- and SST-expressing cells, in two models of acute chemoconvulsive seizure initiation in vivo. Chemoconvulsant models are reliable and consistent across animals and provide a window into the transition from healthy to pathological neural activity, separating interneuron activity and inhibitory influence in the initial transition to ictal activity from alterations in interneuron activity that are known to occur as a result of seizure-induced cell death or sprouting. The inhibitory influence of PV and SST interneuron firing on nearby neurons remains largely intact throughout the preictal and early ictal periods, suggesting that seizure activity does not arise from a failure of GABAergic inhibition from these cells. Instead, PV and SST cells exhibit cell-type-specific differences in their preictal activity and in the evolution of their sensitivity to input. Our findings suggest that the onset of ictal activity is associated with dysregulation of the distinct roles these interneuron populations play in the local hippocampal circuit.

## Results

To examine the preictal activity of identified hippocampal interneurons, we performed tetrode recordings of isolated single units and local field potentials (LFPs) from hippocampal CA1 in lightly anesthetized mice expressing Channelrhodopsin-2 (ChR2) in target cells. Seizure activity was induced with systemic administration of the chemoconvulsant Pentylenetetrazol (PTZ) (*Figure 1A*). During the baseline period, we identified PV- (n = 56 cells in 45 mice) and SST- (n = 42 cells in 34 mice) expressing interneurons in PV-Cre/ChR2 or SST-Cre/ChR2 mice, respectively, by their short-latency, low-jitter responses to blue light (*Figure 1B*; see Materials and methods) (*Cardin et al., 2009*; *Lima et al., 2009*). On average, PV cells displayed narrow spike waveforms, whereas SST cells exhibited broader waveforms (*Figure 1C* and *Figure 1—figure supplement 1A-B*). However, there was extensive overlap of waveform characteristics among the populations (*Figure 1—figure supplement 1A-B*), indicating that spike waveform alone is not sufficient to distinguish cell types under these conditions. Histological analysis confirmed that the two identified interneuron populations were largely nonoverlapping within CA1 (*Figure 1—figure supplement 1C-K*). In some experiments, unidentified cells (n = 49 cells in 26 mice) were simultaneously recorded along with ChR2-tagged units in PV-Cre and SST-Cre mice or in the pyramidal cell layer of wild-type mice (*Figure 1—figure supplement 1A-B*). A subset of these unidentified cells (n = 26 cells in 16 mice) were regular spiking (RS), putative excitatory cells with characteristic broad spike waveforms and relatively low baseline firing rates (*Figure 1C* and *Figure 1—figure supplement 1*).

In an initial series of experiments, we assessed the spontaneous activity of these three cell classes during a baseline period and four preictal periods leading into PTZ-induced seizure under light anesthesia (*Figure 1D*). For each animal, the preseizure period was divided into four equally spaced

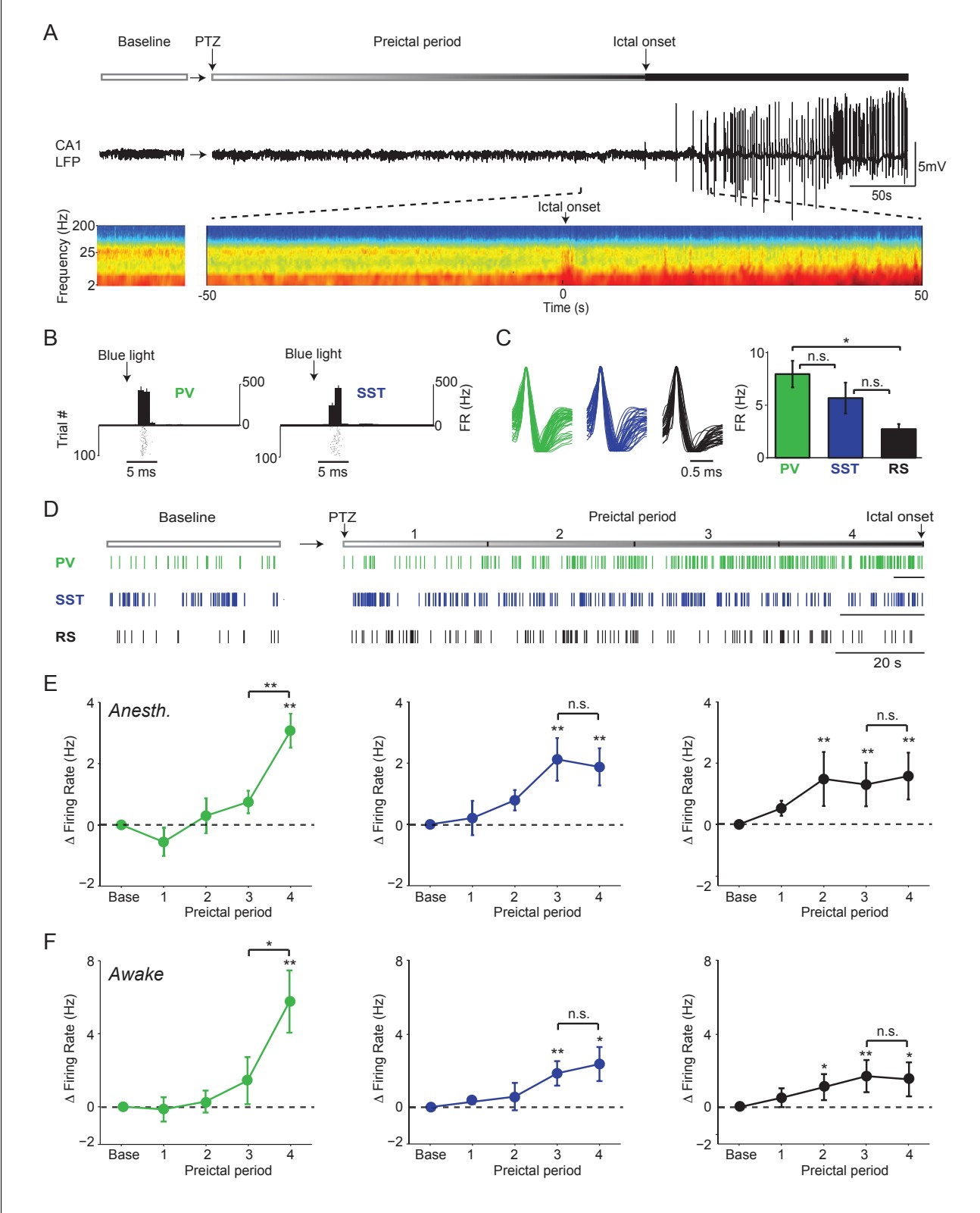

**Figure 1.** Firing rate changes of ChR2-tagged interneurons precede seizure initiation (**A**) Upper: Schematic of experimental paradigm showing baseline, PTZ injection and preictal period leading up to ictal onset. Middle: LFP trace from hippocampal CA1 recording. Lower: Spectrogram showing LFP power as a function of time from ictal onset (x-axis) and frequency (y-axis; shown on log-10 scale). (**B**) Peri-pulse time-histogram around 5 ms laser pulses together with raster plot during baseline for example PV- and SST-expressing interneurons in PV-Cre/ChR2 and SST-Cre/ChR2 mice,

*Figure 1 continued on next page*

*Figure 1 continued*

respectively. (C) Left: Averaged, normalized action potential waveforms for all recorded PV, SST and putative RS cells. Right: Mean baseline firing rates (Hz) for PV, SST and RS cells recorded from hippocampal CA1. (D) Spike trains for example PV, SST and RS cells. All horizontal scale bars correspond to 20 s. Note that cells were recorded from different animals that each had different latencies to ictal onset. (E) Mean changes in firing rate as compared to baseline over four preictal periods for PV (n = 18 cells in 13 mice), SST (n = 16 cells in 10 mice) and RS (n = 20 cells in 12 mice) populations in lightly anesthetized animals. (F) Mean changes in firing rate as compared to baseline over four preictal periods for PV (n = 16 cells in 12 mice), SST (n = 14 cells in 13 mice), and RS (n = 28 cells in 16 mice) populations in awake behaving animals. Note that pairwise statistical comparisons are only shown between 3rd and 4th preictal period. Error bars denote mean ±s.e.m. *$p < 0.0125$, **$p < 0.0025$.
DOI: https://doi.org/10.7554/eLife.40750.002

The following source data and figure supplements are available for figure 1:

**Source data 1.** *Figure 1* - Statistical results.
DOI: https://doi.org/10.7554/eLife.40750.006
**Figure supplement 1.** Spike waveform properties and histological characterization of hippocampal CA1 interneurons.
DOI: https://doi.org/10.7554/eLife.40750.003
**Figure supplement 2.** Spontaneous preictal firing rate trajectories.
DOI: https://doi.org/10.7554/eLife.40750.004
**Figure supplement 3.** Preictal changes in interneuron activity during awake seizure initiation.
DOI: https://doi.org/10.7554/eLife.40750.005

quartiles based on the latency from chemoconvulsant administration to ictal onset. PV, SST and RS cells exhibited increased firing rates following PTZ administration, but showed markedly different firing rate trajectories (*Figure 1E*). Strikingly, we found that most PV cells strongly increased their firing rate in the last preictal period as compared to the first (94.4%, $p < 0.001$, Binomial test) or third (83.3%, $p < 0.001$). In contrast, this sharp, late increase in firing rate was not observed in SST or RS cells (*Figure 1E*). Increased firing of PV, SST, and RS cells was independent of the latency to ictal onset (*Figure 1—figure supplement 2A-E*) and was observed in the absence of significant changes in spike waveform amplitude over time (*Figure 1—figure supplement 2F*). There was no difference in the preictal trajectory of FS and non-FS PV interneurons (*Figure 1—figure supplement 2C*).

To identify common elements across different models of seizure initiation, we compared our PTZ data with recordings from identified interneurons during acute Pilocarpine-induced seizures. We observed a similar increase in PV cell firing rates during the late preictal period preceding Pilocarpine-induced seizures (*Figure 1—figure supplement 2G-J*), suggesting that this is not a unique feature of PTZ-induced seizures. As in the PTZ model, there was no difference in the preictal trajectory of FS and non-FS PV interneurons (*Figure 1—figure supplement 2K*).

In a separate series of experiments, we recorded from identified PV (n = 12 cells in eight mice), SST (n = 10 cells in nine mice), and RS (n = 28 cells in 14 mice) cells in awake head-fixed mice during PTZ seizure initiation (*Figure 1F*, *Figure 1—figure supplement 3*). We found firing rate trajectories similar to those observed in the anesthetized paradigm for all cell classes. Both PV and SST interneuron firing increased during the preictal period, as did RS cell firing. However, only PV interneurons showed a sharp increase in firing rate immediately prior to seizure onset (*Figure 1F*).

We next examined whether interneuron firing remained elevated during the ictal period. Rigorous spike waveform identification after ictal onset is highly challenging. However, we were able to track a subset of recorded neurons through an initial 60 s ictal period (*Figure 2A*). We found that after ictal onset, PV and SST firing decreased from the preictal peak back to baseline levels (*Figure 2B–C*). In contrast, RS cells continued to fire at elevated rates (*Figure 2D*), suggesting a sustained early ictal increase in the activity of excitatory, but not inhibitory, neurons.

To assay whether the observed changes in the activity of interneurons were associated with altered inhibition of their targets, we measured the impact of ChR2-evoked interneuron spiking on the firing rate of nearby RS cells. During baseline activity, we found that the firing rate of RS cells decreased following ChR2-evoked PV and SST cell spiking (*Figure 3A–B*). We compared the impact of ChR2-evoked inhibition during the four preictal periods and an additional period immediately following ictal onset. RS firing suppression was not significantly changed across the preictal and early ictal periods as compared to baseline when the PV cells were driven by optogenetic stimulation (see Materials and methods; *Figure 3C–D*, *Figure 3—figure supplement 1*). RS suppression by SST cell spiking was likewise maintained throughout preictal and early ictal periods. We further measured RS

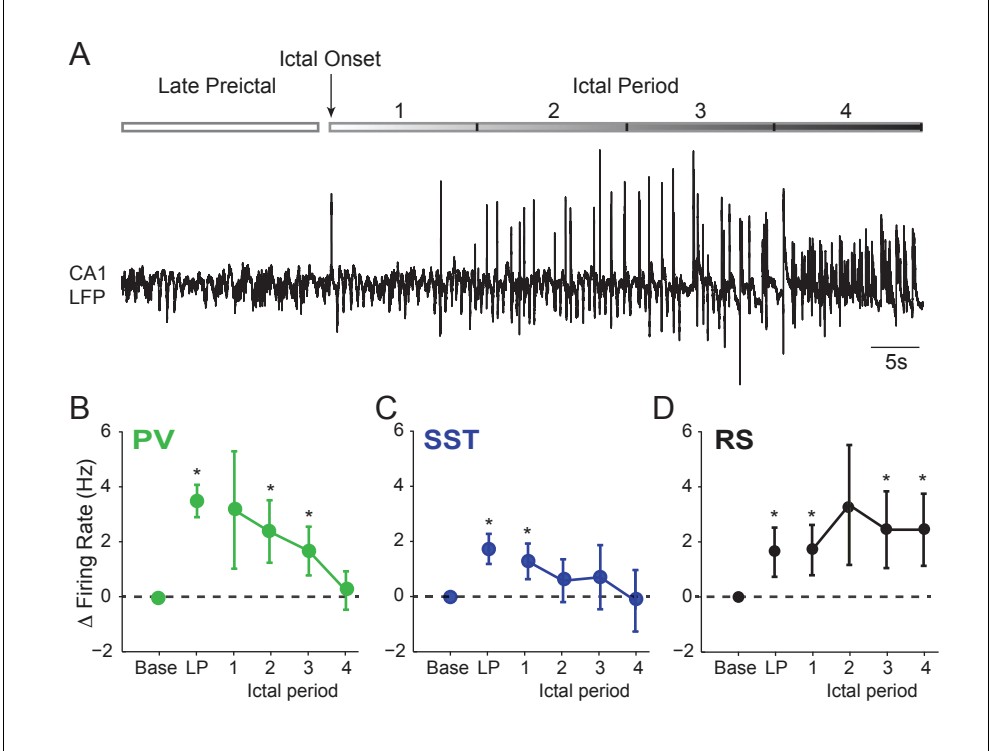

**Figure 2.** Interneuron firing rates decrease after ictal onset. (**A**) Upper: Schematic of experimental paradigm showing the late preictal (LP) period, ictal onset timepoint, and ictal period. Lower: LFP trace from hippocampal CA1 recording. (**B**) Mean changes in firing rate as compared to baseline during four ictal periods for PV interneurons (n = 9 cells in eight mice). Firing rate in fourth ictal period was significantly lower than in the LP period for PV cells. (**C**) Mean ictal changes in SST interneuron firing rate (n = 18 cells in 17 mice). (**D**) Mean ictal changes in RS cell firing rate (n = 16 cells in nine mice). Note, LP values are taken from the 4th preictal period shown in *Figure 1*. All statistical comparisons are to the baseline period. Error bars denote mean ±s.e.m. *p<0.0125, **p<0.0025.

DOI: https://doi.org/10.7554/eLife.40750.007

The following source data is available for figure 2:

**Source data 1.** *Figure 2* - Statistical results.

DOI: https://doi.org/10.7554/eLife.40750.008

suppression in awake animals during seizure initiation. We were not able to maintain high-quality recordings after ictal onset in awake animals. However, RS suppression in response to PV and SST spiking was consistently observed during the preictal period of awake seizure initiation (*Figure 3D*).

We next explored whether preictal changes in interneuron activity levels were accompanied by changes in the temporal spike pattern. Immediately preceding ictal onset in both anesthetized and awake animals, PV, but not SST or RS, cell firing became significantly more regular (i.e., less bursty) (*Figure 4A–B*, *Figure 4—figure supplement 1E*). In addition, PV cells showed an increased tendency to fire spikes separated by short (<10 ms) inter-spike-intervals (ISI; *Figure 4—figure supplement 1*). Unidentified cells with narrow spikes did not exhibit changes in firing rate, firing regularity, or ISI statistics (*Figure 4—figure supplement 1B-D*).

PV and SST interneurons normally exhibit temporally patterned activity and highly precise entrainment to hippocampal gamma and theta rhythms (*Klausberger et al., 2003*; *Klausberger et al., 2005*; *Tukker et al., 2007*; *Diba et al., 2014*). We therefore computed the mean spike field coherence (SFC) for PV, SST and RS cells during baseline activity and across the four preictal periods (*Figure 4C*, *Figure 4—figure supplement 1*). We observed a prominent peak in the 20 – 28 Hz band of the LFP (*Chen et al., 2011*; *Cabral et al., 2014*; *Sauer et al., 2015*) under baseline and preictal conditions (*Figure 1A*). PV cells showed a peak in SFC in the 20 – 28 Hz frequency band that

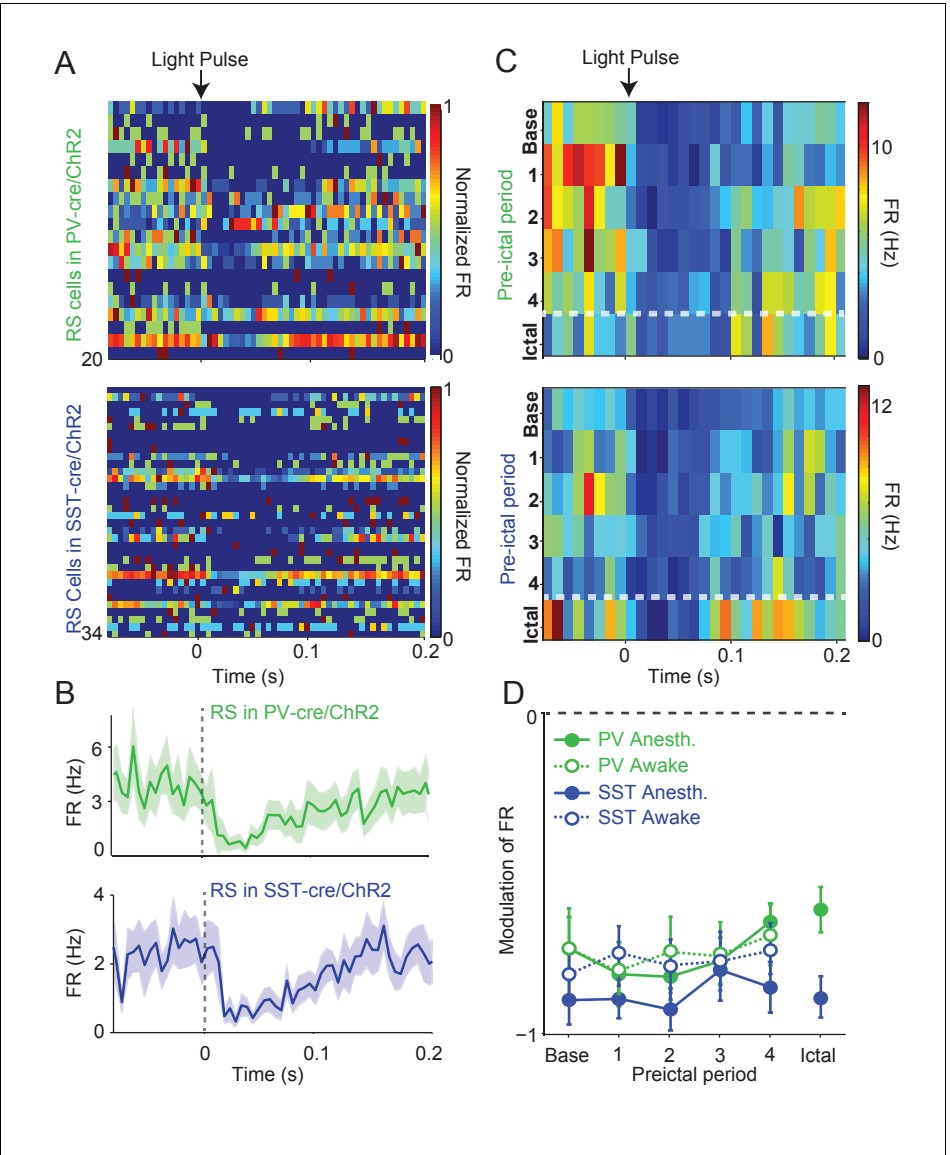

**Figure 3.** Intact preictal suppression of RS firing by evoked inhibition. (**A**) Normalized firing rate (FR) as a function of time (**s**) from laser pulse onset (t = 0) for RS cells recorded simultaneously with PV cells in PV-Cre/ChR2 mice (n = 20 cells in 12 mice) or recorded with SST cells in SST-Cre/ChR2 mice (n = 34 cells in 16 mice) during baseline activity. For purposes of visualization, firing rates were normalized to the maximum firing rate for each cell. (**B**) Mean changes in baseline firing rate (Hz) of RS cells as a function of time (**s**) from laser pulse onset (dashed grey line). (**C**) Upper: Average firing rate of RS cells (n = 11 cells in eight mice) recorded during PV/ChR2 experiments as a function of time (**s**) from laser pulse onset during baseline, the four preictal periods and a 60 s period following ictal onset after PTZ administration (see Materials and methods). Lower: Average firing rate of RS cells (n = 11 cells in seven mice) recorded during SST/ChR2 experiments as a function of time (**s**) from laser pulse onset during baseline, the four preictal periods and following ictal onset. For these plots, we considered only light pulses of high light intensity (see Materials and methods). (**D**). Mean modulation of firing rate (see Materials and methods) after high-intensity laser pulse (0 to 50 ms) compared to pre-pulse (−200 to 0 ms) firing rate for RS cells recorded during PV/ChR2 (green) and SST/ChR2 (blue) experiments. Data from lightly anesthetized animals are shown as filled symbols and data from awake animals are shown as open symbols. Modulation scores for RS in PV-Cre and RS in SST-Cre were significantly different from zero for all periods. Error bars denote mean ±s.e.m.

DOI: https://doi.org/10.7554/eLife.40750.009

The following source data and figure supplement are available for figure 3:

**Source data 1.** *Figure 3* - Statistical results.

*Figure 3 continued on next page*

*Figure 3 continued*

DOI: https://doi.org/10.7554/eLife.40750.011

**Figure supplement 1.** Modulation of RS firing by medium intensity stimulation of local inhibitory interneurons.

DOI: https://doi.org/10.7554/eLife.40750.010

decreased significantly during the preictal period (*Figure 4C–D*) in the absence of any loss in LFP power or changes in SFC in the theta or high gamma bands, other prominent hippocampal rhythms associated with interneuron activity (*Figure 4—figure supplement 1F-H*). In contrast, neither SST nor RS cells showed a change in SFC across the preictal periods, suggesting a specific decoupling of

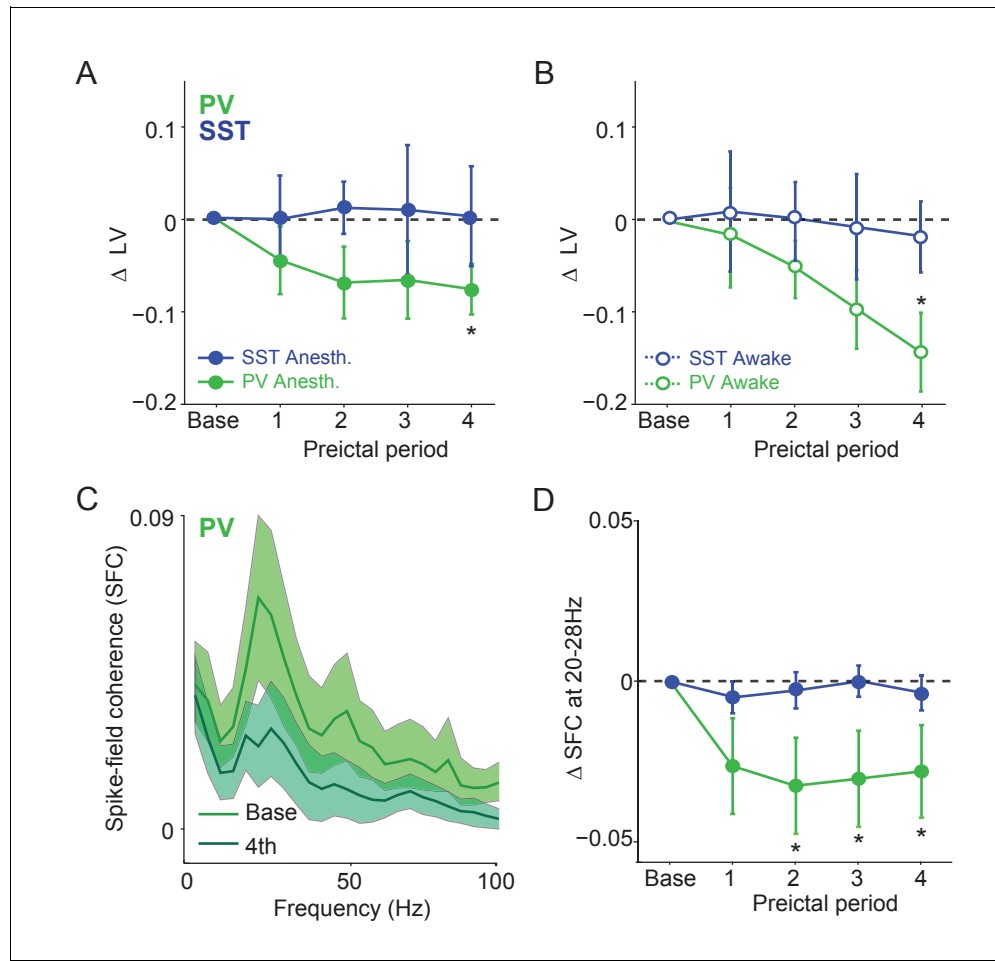

**Figure 4.** Preictal changes in temporal patterning of interneuron activity (**A**) Mean changes in local variation (LV) of firing as compared to baseline over four preictal periods for PV and SST cells during acute PTZ seizure in lightly anesthetized animals. LV is a measure of firing irregularity, where decreases in LV indicate more regular firing (see Materials and methods). (**B**) Same as A, but for interneurons recorded in awake animals. (**C**) Mean spike-field coherence (SFC) as a function of frequency (Hz) during baseline and fourth preictal period for PV-expressing cells. (**D**) Mean changes in SFC in 20 – 28 Hz band as compared to baseline over four preictal periods for PV and SST cells. Error bars denote mean ±s.e.m. *p<0.0125.

DOI: https://doi.org/10.7554/eLife.40750.012

The following source data and figure supplement are available for figure 4:

**Source data 1.** *Figure 4* - Statistical results.

DOI: https://doi.org/10.7554/eLife.40750.014

**Figure supplement 1.** Preictal changes in the temporal pattern of interneuron activity.

DOI: https://doi.org/10.7554/eLife.40750.013

PV cells from their normal temporal relationship with the local hippocampal network during the onset of ictal activity (*Figure 4—figure supplement 1F-H*).

To examine whether preictal changes in interneuron output were accompanied by changes in sensitivity to input, we tested the responses of PV and SST cells to optogenetic stimulation during each preictal period. We measured the probability of interneuron spiking in response to light pulses of varying intensity (*Figure 5A*). PV cells showed no progressive change in spike probability over the preictal period (*Figure 5B–C*). In contrast, SST cells showed an increase in the slope of their response curve and a decrease in their maximal response probability (Rmax) across the preictal periods. Changes in interneuron sensitivity to input were not affected by seizure latency (*Figure 5—figure supplement 1*). Together, these data suggest a progressive preictal attenuation of the response of SST cells to input, potentially limiting their recruitment by network inputs.

## Discussion

During spontaneous and evoked neural activity, excitation and inhibition are tightly coupled in amplitude (*Shu et al., 2003*; *Haider et al., 2006*; *Xue et al., 2014*) and temporal pattern (*Pouille and Scanziani, 2001*; *Wehr and Zador, 2003*; *Higley and Contreras, 2006*; *Cardin et al., 2010*). Disruption of the excitatory-inhibitory balance profoundly alters circuit function (*Fritschy, 2008*; *Žiburkus et al., 2013*) and is hypothesized to play a significant role in the pathophysiological patterns of brain activity leading to the onset of seizure. The transition to ictal activity has been characterized in a variety of ways in rodents, including by manual validation (*Henderson et al., 2014*), threshold detection of increased EEG power (*Khoshkhoo et al., 2017*), and combined measurement of EEG power trajectory and spectral features (*Krook-Magnuson et al., 2014*). However, the patterns of activity underlying these transitions from stable to epileptiform activity are not well understood.

In the current study, where we utilized an acute chemoconvulsant seizure induction, the onset of the ictal period was defined quantitatively by increased LFP amplitude, which largely corresponded to the first ictal spike. These early events occurred in advance of full ictal recruitment. We found that PV- and SST-expressing interneurons and RS, putative excitatory cells exhibited different preictal firing rate trajectories. SST and RS cells showed an early, modest increase in firing, whereas >90% of PV cells exhibited a late increase in firing immediately before ictal onset. During the early ictal period, PV and SST cell firing rates returned to baseline levels, whereas RS cell firing remained elevated. These changes in PV and SST cell firing rates occurred in the absence of changes in the impact of ChR2-evoked inhibition on local RS cells. Overall, these results suggest that the initial changes in network activity leading to early stages of seizure initiation may not be associated either with an overall failure of GABAergic inhibition from PV or SST cells or with unconstrained excitation. However, we note that our findings do not preclude additional impairments in inhibitory function at the time of ictal recruitment or later in the ictal period.

Fast-spiking, PV-expressing interneurons are coupled to both high-frequency and theta rhythms in the hippocampus, and are thought to contribute to maintaining the fine temporal organization of excitatory-inhibitory interactions (*Buzsáki et al., 2003*; *Csicsvari et al., 2003*; *Klausberger et al., 2003*; *Diba et al., 2014*; *Amilhon et al., 2015*; *Forro et al., 2015*; *Huh et al., 2016*). The earliest preictal change we observed was a loss of the strong spike-field coherence of PV cell spiking to a prominent low-gamma band previously observed in mouse hippocampus (*Chen et al., 2011*; *Cabral et al., 2014*; *Sauer et al., 2015*). Previous work has suggested that increased inhibitory synaptic activity or the onset of depolarization block in fast-spiking interneurons could precipitate seizure onset (*Velazquez and Carlen, 1999*; *Timofeev et al., 2002*; *Ziburkus et al., 2006*; *Fujiwara-Tsukamoto et al., 2007*; *Gnatkovsky et al., 2008*; *Cammarota et al., 2013*; *Kim and Nykamp, 2017*), and we found that the onset of ictal spikes occurred immediately after a sharp increase in PV interneuron spiking in both anesthetized and awake animals. However, we found no evidence for decreased interneuron spike amplitude prior to ictal onset, suggesting that these interneurons did not enter depolarization block prior to ictal onset. Late preictal increases in PV interneuron firing could lead to irregular firing patterns or bursts in excitatory neurons, potentially resulting in hyper-synchronized, pathological entrainment of excitatory activity (*Shiri et al., 2015*; *Yekhlef et al., 2015*; *Avoli et al., 2016*).

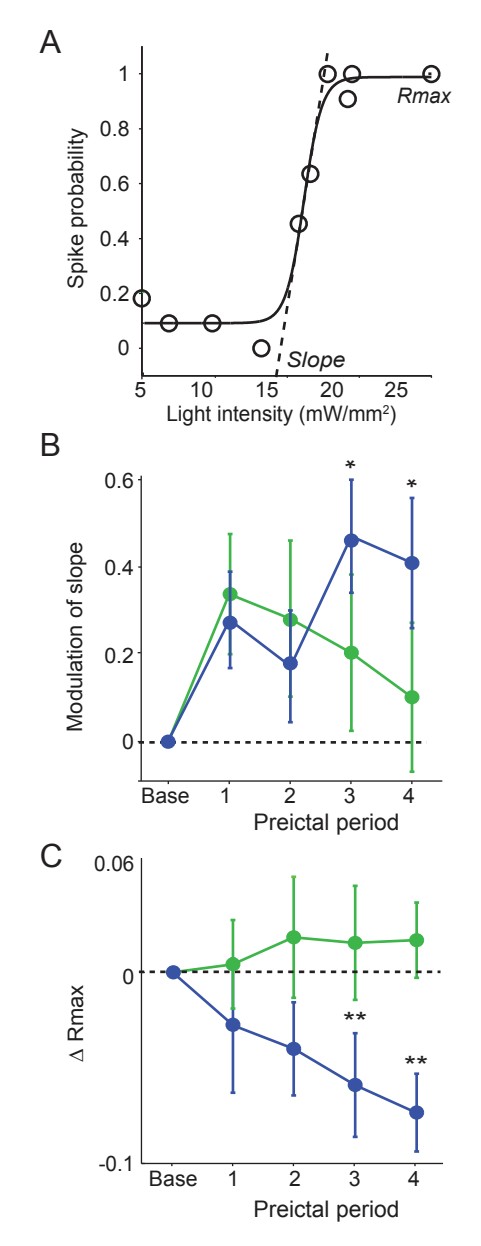

**Figure 5.** Preictal changes in interneuron sensitivity to inputs (**A**) Example response curve of spike probability as a function of light pulse intensity (mW/mm$^2$) with sigmoid curve fit, showing slope and maximal response probability (Rmax). (**B**) Mean modulation of the slope parameter compared to baseline over four preictal periods for PV (n = 17 cells in 15 mice) and SST (n = 21 cells in 20 mice) cells. (**C**) Mean change in Rmax compared to baseline (dotted line) over four preictal periods for PV and SST cells. (**B–C**) Significance to baseline: *p<0.0125, **p<0.0025, ***p<0.00025. Error bars denote mean ±s.e.m.

DOI: https://doi.org/10.7554/eLife.40750.015

The following source data and figure supplement are available for figure 5:

*Figure 5 continued on next page*

In contrast to the PV interneurons, SST interneurons showed a sustained elevation of preictal firing without a late increase in firing rate preceding the transition to ictal activity. SST interneurons did not show a change in the overall temporal pattern of their spiking or entrainment to specific hippocampal rhythms. However, optogenetic stimulation revealed that SST, but not PV, cells exhibited altered sensitivity to input and a decreased maximum response to inputs during the late preictal period. OLM cells, the primary group of SST neurons targeted in our experiments, receive input from CA1, but not CA3 or the entorhinal cortex (*Blasco-Ibáñez and Freund, 1995*). They innervate the distal tuft dendrites of local excitatory pyramidal neurons, providing potent recurrent feedback within the local circuit (*Bloss et al., 2016*). SST inhibition restrains the occurrence of hippocampal pyramidal neuron bursts, which promote highly effective transmission to downstream targets (*Lisman, 1997*; *Royer et al., 2012*), and may regulate plateau potentials and plasticity in pyramidal cell dendrites (*Takahashi and Magee, 2009*; *Lovett-Barron et al., 2012*). SST cells show a preictal decrease in dynamic range and loss of ability to respond to depolarizing inputs, potentially leading to nonlinear amplification of excitatory activity in the local hippocampal circuit.

Previous work has found extensive heterogeneity in preictal firing rate trajectories of extracellularly recorded putative excitatory and inhibitory neurons. Intracellular and extracellular recordings of fast-spiking, putative PV interneurons have found increased preictal firing rates in cortical networks (*Timofeev et al., 2002*; *Gnatkovsky et al., 2008*) or a transient increase in firing immediately preceding ictal onset for some hippocampal cells (*Grasse et al., 2013*; *Toyoda et al., 2015*). To our knowledge, there are no previous data on the preictal firing rate trajectories of SST interneurons, although recent work in motor cortex found general population recruitment of both PV and SST interneurons following contralateral seizure onset (*Khoshkhoo et al., 2017*). In turn, some reports suggest increased preictal firing of regular spiking putative excitatory neurons in hippocampus (*Cymerblit-Sabba and Schiller, 2010*; *Jiruska et al., 2010*; *Fujita et al., 2014*), but others find a mix of increases and decreases within multiple hippocampal areas (*Bower and Buckmaster, 2008*; *Toyoda et al., 2015*). Although the findings of a number of studies in both neocortex and hippocampus are thus consistent with inhibitory disruption, it also remains

*Figure 5 continued*

**Source data 1.** *Figure 5* - Statistical results.
DOI: https://doi.org/10.7554/eLife.40750.017
**Figure supplement 1.** Data for ChR2 stimulation reported in *Figure 5* were divided into short (<300 s) and long (>300 s) groups according to preictal latency (See *Figure 1—figure supplement 2*).
DOI: https://doi.org/10.7554/eLife.40750.016

possible that the mechanisms underlying seizures in these areas are different.

We found that identification by action potential waveform or baseline firing rate was not a reliable indicator of cell identity for hippocampal PV interneurons. In addition, in agreement with previous work (*Klausberger et al., 2003*; *Halabisky et al., 2006*; *Katona et al., 2014*), we found that SST interneurons exhibited varied action potential durations and could not be distinguished from regular spiking excitatory neurons by waveform. Optical tagging with Cre-dependent ChR2 allowed identification of each population despite overlapping action potential characteristics, and we found that the three cell classes we examined demonstrated distinct trajectories. Indeed, >90% of PV interneurons exhibited a characteristic increase in firing immediately prior to the first ictal spike, suggesting that some of the heterogeneity observed in previous studies may arise from a mixed population of unidentified cells.

We examined interneuron activity in two chemoconvulsant models of acute induction of status epilepticus. These models provide insight into the mechanisms by which normal, healthy neural circuits transition to pathological patterns of activation, although the precise commonality of circuit-level mechanisms underlying induced and spontaneous seizures has not been fully explored. We used both PTZ, thought to be a competitive antagonist of the GABA$_A$ receptor (*Huang et al., 2001*), and Pilocarpine, a nonselective muscarinic acetylcholine receptor agonist (*Turski et al., 1989*). Despite distinct pharmacological mechanisms, we found similar trajectories for PV interneuron activity in both models, as well as in awake animals, suggesting that preictal increases in PV activity may be a common element of acute seizure initiation. However, the exact timing of interneuron activity varied slightly across models. Because these drugs arrive rapidly in the brain at effective concentrations following systemic administration (*Yonekawa et al., 1980*; *Ramzan and Levy, 1985*), it is unlikely that the progression of interneuron firing changes resulted from gradual accumulation of chemoconvulsants in neural tissue. Our experiments were largely conducted under anesthesia, which could potentially prolong the preictal period and reduce seizure activity (*Murao et al., 2002*; *Fang and Wang, 2015*; *Grover et al., 2016*). However, the light anesthesia we used allowed normal spontaneous hippocampal firing patterns to be maintained and promoted short ictal onset delays similar to those in awake animals. Furthermore, we observed similar PV and SST trajectories in awake and anesthetized animals.

One potential function of inhibition during seizure is the restraint of ictal activity. The spreading ictal wavefront generates feedforward excitation, which recruits powerful feedforward inhibition that may restrict the postsynaptic recruitment of spiking activity. Synaptic inhibition restrains the spread of ictal activity in the cortex (*Trevelyan et al., 2006*; *Trevelyan et al., 2007*; *Schevon et al., 2012*; *Sessolo et al., 2015*), and surround inhibition around epileptic foci has been observed in animal and human seizures (*Prince and Wilder, 1967*; *Schwartz and Bonhoeffer, 2001*; *Timofeev et al., 2004*; *Trevelyan et al., 2006*; *Trevelyan et al., 2007*). We observed preictal increases in both PV and SST firing rates, raising the possibility that compensatory increases in inhibition transiently restrain oncoming ictal events. However, the increasing loss of normal temporal structure and input sensitivity by the PV and SST interneuron populations, respectively, may contribute to the transition to a pathological state.

Overall, our data suggest that cell-type-specific disruption of finely tuned interneuron relationships with the local hippocampal circuit may contribute to reorganization of inhibitory activity prior to seizure initiation. These findings highlight the complex involvement of distinct GABAergic interneuron populations in early stages of pathological activity in the hippocampal circuit. Indeed, although the PV and SST populations encompass a large proportion of hippocampal interneurons, other inhibitory cell types may be differentially engaged by seizure initiation and remain to be investigated.

# Materials and methods

## Animals

All experiments were approved by the Institutional Animal Care and Use Committee of Yale University. Experiments were performed using 2- to 6-month-old male and female mice heterozygous for PV-Cre (Jackson Laboratory strain #008069; RRID:IMSR_JAX:008069) or SST-Cre (Jackson Laboratory strain #013044; RRID:IMSR_JAX:013044). All mice were heterozygous for Ai32 (Jackson Laboratory strain #012569; RRID:IMSR_JAX:012059), which expresses a channelrhodopsin-2/EYFP fusion protein in Cre recombinase containing cells (*Hippenmeyer et al., 2005*; *Taniguchi et al., 2011*; *Madisen et al., 2012*). For immunohistological verification of the fidelity of the two Cre lines, they were also crossed to the tdTomato-expressing Ai9 line (Jackson Laboratory strain #007909; RRID: IMSR_JAX:007909)(*Madisen et al., 2010*).

## Histology

PV-Cre/Ai9 and SST-Cre/Ai9 reporter mice were anesthetized with isoflurane and then transcardially perfused with 0.9% saline solution, followed by 4% paraformaldehyde in 0.1 M phosphate buffer (PB). The brains were post-fixed in the same solution at room temperature for 45 mins and cryoprotected by immersion in 15% and 30% sucrose solutions in PB at 4°C until they sank. Coronal sections (20 μm) from hippocampal CA1 were obtained from each brain using a cryostat (Leica) at −20°C. The sections were washed and blocked against non-specific antibody binding with 2% normal goat serum in PB containing 0.1% Triton X-100 (PB-TX) for 1 hr at room temperature and then incubated with either monoclonal PV (monoclonal; 1:500; Sigma P3088; RRID:AB_477329) or SST (monoclonal; 1:200; Millipore MAB354; RRID:AB_2255365) antibody overnight at 4°C. All sections were again washed in PB-TX and then incubated in an Alexa 488 secondary antibody (1:1000; Invitrogen) for 1 hr at room temperature. Finally, the sections were rinsed in PB and DAPI was added before slides were cover-slipped.

Immunostaining and counting were performed on a minimum of three coronal sections from at least three PV-Cre/Ai9 or SST-Cre/Ai9 animals for each respective condition. Hippocampal analyses were carried out in CA1 and ImageJ (Wayne Rasband, NIH) was used for image processing and counting. To minimize counting bias, we compared sections of equivalent bregma positions (from −1.5 mm to −2.0 mm relative to bregma), defined according to the Mouse Brain atlas (Franklin and Paxinos, 2001). For comparative purposes, we also analyzed a series of caudal sections of equivalent bregma positions (from −2.7 to −3.2 mm relative to bregma; *Figure 1—figure supplement 1H–K*).

## Acute seizure induction protocol

Most recordings were performed in mice anesthetized with. 02 ml ketamine (80 mg/kg)/xylazine (5 m/kg) administered by intraperitoneal (IP) injection before they were intubated, mechanically ventilated and then paralyzed using. 024 mg/g gallamine triethiodide (Sigma). Animals were then fixed in place with a headpost. A 200 μm optical fiber was lowered through the cortex until it was ~200 μm above the hippocampal CA1. Recording electrodes were then lowered into CA1 and the optical fiber position was adjusted in small steps. Simultaneous recordings of isolated single units and LFPs were made during spontaneous baseline activity, when ChR2-expressing PV or SST interneurons were identified with short pulses of light at 473 nm, and during periods following chemoconvulsant administration. The preictal period was defined as the time from chemoconvulsant injection until the onset of ictal activity. In all experiments, seizures were induced pharmacologically with IP injection of either 120 mg/kg PTZ, a GABA-A receptor antagonist, or 500 mg/kg Pilocarpine, a muscarinic AChR agonist. There was no significant difference in seizure latency between PV-Cre/ChR2 and SST-Cre/ChR2 animals.

## Headpost surgery and wheel training

For recordings performed in awake animals, mice were initially handled for 5 – 10 min/day for 5 days prior to a headpost surgery. On the day of the surgery, the mouse was anesthetized with isoflurane and the scalp was shaved and cleaned three times with Betadine solution. An incision was made at the midline and the scalp resected to each side to leave an open area of skull. Two skull screws (McMaster-Carr) were placed at the anterior and posterior poles. Two nuts (McMaster-Carr) were

glued in place over the bregma point with cyanoacrylate and secured with C&B-Metabond (Butler Schein). The Metabond was extended along the sides and back of the skull to cover each screw, leaving a bilateral window of skull uncovered over primary visual cortex. The exposed skull was covered by a layer of cyanoacrylate. The skin was then glued to the edge of the Metabond with cyanoacrylate. Analgesics were given immediately after the surgery and on the two following days to aid recovery. Mice were given a course of antibiotics (Sulfatrim, Butler Schein) to prevent infection and were allowed to recover for 3 – 5 days following implant surgery before beginning wheel training.

Once recovered from the surgery, mice were trained with a headpost on the wheel apparatus. The mouse wheel apparatus was 3D-printed (Shapeways Inc.) in plastic with a 15 cm diameter and integrated axle and was spring-mounted on a fixed base. A programmable magnetic angle sensor (Digikey) was attached for continuous monitoring of wheel motion. Headposts were custom-designed to mimic the natural head angle of the running mouse, and mice were mounted with the center of the body at the apex of the wheel. On each training day, a headpost was attached to the implanted nuts with two screws (McMaster-Carr). The headpost was then secured with thumb screws at two points on the wheel. Mice were headposted in place for increasing intervals on each successive day. If signs of anxiety or distress were noted, the mouse was removed from the headpost and the training interval was not lengthened on the next day. Mice were trained on the wheel for up to 7 days or until they exhibited robust bouts of running activity during each session. Mice that continued to exhibit signs of distress were not used for awake electrophysiology sessions.

## Awake seizure induction protocol

For experiments performed in awake animals, mice were placed on the wheel and fixed in place by the headpost at the beginning of the session. After baseline recordings were made, the mouse was transiently anesthetized for ~30 s with 1% isoflurane without disturbing the electrodes and an IP injection of 120 mg/kg PTZ was administered. The isoflurane was immediately removed and the animal allowed to resume normal waking activity.

## Extracellular recordings

All extracellular single-unit, multi-unit, and LFP recordings were made with custom-designed, moveable arrays of tetrodes manufactured in the lab from Formvar-coated tungsten wire (12.5 μm diameter; California Fine Wire, Grover Beach CA). Tetrodes were targeted to the CA1 field of the dorsal hippocampus (AP:+1.5 – 2 mm; ML: 1.2 – 1.75, Franklin and Paxinos, 2001). Signals were digitized and recorded with a DigitalLynx 4SX system (Neuralynx, Bozeman MT). All data were sampled at 40 kHz and recordings were referenced to the cerebellum. LFP data were recorded with a bandpass 0.1 – 9000 Hz filter and single-unit data was bandpass filtered between 600 – 9000 Hz. All data was analyzed using the Mathworks Fieldtrip toolbox (in particular the Spike toolbox) and Matlab (The Mathworks, Natick, MA) and Igor (WaveMetrics, Lake Oswego, OR) scripts (M. Miri and M. Vinck).

## Spike sorting

Spikes were clustered using previously published methods (*Vinck et al., 2015*; *Batista-Brito et al., 2017*). We first used the KlustaKwik 3.0 software (Kadir, 2013) to identify a maximum of 30 clusters using the waveform energy and energy of the waveform's first derivative as clustering features. We then used a modified version of the M-Clust environment to manually separate units. Units were accepted if a clear separation of the cell relative to all the other noise clusters was observed, which generally was the case when isolation distance (ID) (Schmitzer-Torbert et al., 2005) exceeded 20 (*Vinck et al., 2015*). We further ensured that maximum contamination of the ISI (Inter-spike-interval) histogram did not exceed 0.1% at 1.5 ms. To analyze the firing rates of cells after ictal onset, we manually determined the last point at which cluster separation from the noise was clearly visible. Only cells whose ictal activity could be tracked for at least 60 s were included in the ictal period analysis.

## Analysis of waveform parameters

For each isolated single unit, we computed an average spike waveform for all channels of a tetrode. The waveforms were manually inspected and the channel with the largest peak-to-trough amplitude was used to measure the peak-to-trough duration values as well as mean spike amplitude (*Figure 1—*

*figure supplement 1*). We also computed the repolarization value of the normalized (between −1 and +1) waveforms at 0.9 ms (similar to *Vinck et al., 2015*). Non-light-driven units were categorized as RS cells by their broad waveform, defined as a repolarization value at 0.9 ms smaller than −0.35.

## Firing rate and bursting

The mean firing rate per analysis period was computed as the number of spikes in that period divided by the duration of that period in seconds. Changes in the temporal patterning of preictal firing were detected using two metrics. First, we quantified the propensity to engage in irregular burst firing using the coefficient of local variation (LV; *Figure 2A*), which has been shown to be robust against non-stationarities in firing rates. LV values greater than one indicate irregular firing, whereas LV values smaller than one indicate sub-Poisson regular firing (Shimokawa and Shinomoto, 2009). Second, we computed the log fraction of ISIs between 2 and 10 ms over the fraction of ISIs between 10 and 100 ms, that is, $\mathrm{Log}(\mathrm{ISI}_{short}/\mathrm{ISI}_{long})$, as in *Vinck et al. (2015)*.

## Spike-field locking and LFP power

Coherence is a nonparametric spectral estimate of the frequency-by-frequency linear dependence between two time series. Spike-field coherence, or phase-locking, is a measure of the rhythmic synchrony between the LFP signal in a particular frequency band and the spikes of an individual neuron. Spike-field locking (*Figure 4C–D*) was computed using the Pairwise Phase Consistency, a measure of phase consistency that is not biased by the firing rate or the number of spikes (Vinck et al., 2012). Spike-LFP phases were computed for each spike and frequency separately by computing the Discrete Fourier Transform of Hanning-tapered LFP segments. These segments had a duration of 7 cycles per frequency. LFP power (*Figure 4—figure supplement 1*) was computed by dividing the signal into 1 s segments and computing the DFT with a Hanning taper. We then averaged the LFP power in the 20 – 28 Hz band.

## Optogenetic manipulations

Light activation of ChR2-expressing cells was performed using a 473 nm laser (OptoEngine LLC, Midvale UT). To avoid heating of the brain, we calibrated the light power (<75 mW/mm2) during ChR2 unit tracking experiments in order to ensure a mean spike probability of ~1 spikes per 5 ms light pulse in the targeted population. Real-time output power for each laser was monitored using a photodiode and recorded continuously during the experiment. During baseline periods, we identified ChR2-expressing interneurons using short (5 ms) pulses of blue light, relying on the short latency of ChR2-evoked spikes and the high degree of temporal precision of the evoked spikes. In a subset of experiments (*Figure 5*), we measured the input-output function of ChR2-identified interneurons in response to a calibrated range of light intensities. Due to the potential for manipulation of interneurons to affect the firing of surrounding cells, data from optogenetic manipulation experiments was not used to calculate preictal and ictal firing rates or patterns (*Figures 1–2* and *4*).

## Seizure detection

Ictal onset was identified by examining hippocampal CA1 LFP recordings, and was defined as the first occurrence of an ictal spike following injection of the chemoconvulsant. This generally corresponded to the LFP trace crossing an absolute z-score value >5 as compared to baseline. Sustained, elevated z-scores were generally observed after ictal onset, and ictal onset was typically coincident with the first ictal spike. We used spectrograms to validate that there were no consistent LFP changes prior to ictal onset (*Figure 1A*). Spectrograms of LFP power around ictal onset were computed using a wavelet transform with seven cycles for each frequency and a Hanning taper. LFP power was normalized by dividing by the summed power across the entire trace and taking the base-10 logarithm.

## Definition of analysis periods

Seizure latency varied across mice (*Figure 1—figure supplement 2*). For each experiment, the preictal period from injection to ictal onset was therefore divided into four equal periods. To characterize the progressive changes in hippocampal CA1 activity (*Figures 1–2*), we computed the change in firing activity parameters as compared to baseline for the four preictal periods. For the analysis of

spontaneous activity, we only used baseline and preictal periods that did not contain epochs of laser pulses and selected cells with baseline firing rates greater than 0.1 Hz. For the analyses of evoked activity in *Figures 3* and *5*, all cells were used. For the analysis of RS suppression by ChR2-evoked inhibition, we defined an additional early ictal period as the 60 s following ictal onset. For analysis of ictal spiking, we used only the subset of interneurons whose ictal spike activity could be resolved and we divided the 60 s ictal period into four periods of 15 s each.

Analysis of preictal activity in awake mice was adjusted to account for intermittent bouts of walking and running, which are known to regulate excitatory and inhibitory activity in the hippocampus (*Komisaruk and Olds, 1968*; *Buzsáki et al., 1983*; *Fox et al., 1986*; *Geisler et al., 2007*). We therefore used a previously validated change-point detection algorithm to detect periods of quiescence and locomotion on the wheel (*Vinck et al., 2015*; *Batista-Brito et al., 2017*). Firing rate analysis was restricted to periods of quiescence within these epochs to exclude the effects of locomotion on firing rates, and the preictal portion of the recording was divided into four preictal periods as for the anesthetized data.

## Response curves

In a subset of experiments (*Figure 5*), we measured the input-output function of ChR2-identified interneurons in response to a calibrated range of light intensities. We then made a sigmoid fit of the probability of spiking as a function of light intensity. This sigmoid fit was defined as

$$p(I) = B + \frac{S}{1+\exp\left(-\frac{I-c}{A}\right)}$$

here, $p(I)$ is the fitted probability of a spike in the 2 – 15 ms following laser pulse onset, $I$ is the laser intensity, $S$ is a scaling factor, $c$ is the c50, and $1/A$ is the slope. The Rmax was defined as the value of $p(I)$ at the maximum laser intensity tested. We fitted these curves by minimizing the absolute deviation between fit and data (i.e., the L1 norm) using MATLAB's *fminsearch* function. To avoid finding a local minimum, we randomly chose 64 different starting values for the different parameters and selected the fit that minimized the error across all 64 initializations.

## RS cell inhibition

During a subset of our recordings in PV-Cre/ChR2 and SST-Cre/ChR2 mice, we simultaneously recorded the activity of local RS cells, defined as described above, and monitored changes in preictal and early ictal inhibition of these units. To quantify the extent to which RS cells were inhibited following the light pulses, we compared their firing rates in the 50 ms period post-light pulses ($FR_{post}$) with their firing rate in the 200 ms prior to light pulses ($FR_{pre}$). We then computed the modulation of RS firing rates (*Figure 3C–D*) as

$$y = [FR_{post} - FR_{pre}]/[FR_{post} + FR_{pre}].$$

We computed this modulation separately for pulses of medium and high intensity (*Figure 3—figure supplement 1*). The medium intensity level was defined as the level at which the simultaneously driven PV cells had, on average, a 50% firing probability. The highest intensity level was the level at which the simultaneously recorded PV cell spiking reached its maximum spike probability.

## Statistical testing

Paired and unpaired Rank Wilcoxon and Kruskal Wallis tests were used throughout the manuscript to avoid the assumptions made by parametric statistical tests. $\alpha$ values were adjusted in cases where multiple comparisons were performed on the same data set, and all measures of significance in these cases are reported as Bonferroni corrected values. Exact p values are reported in the Source Data files for each figure.

## Acknowledgements

The authors thank M Higley for comments on the manuscript, H Blumenfeld for discussion of seizure models, and the Yale Center for Analytical Science for assistance with statistical analysis. This work was funded by an NSF graduate fellowship and a Ford Foundation graduate fellowship to MLM, a Rubicon (NWO) postdoctoral fellowship and a Human Frontiers postdoctoral fellowship to MV, and a Klingenstein fellowship, a Whitehall grant, an Alfred P Sloan Fellowship, NIH grant R01 EY022951,

and a grant from the Swebilius Foundation to JAC. This work was also supported by NIH grant P30 EY026878.

## Additional information

### Funding

| Funder | Grant reference number | Author |
|---|---|---|
| National Science Foundation | Graduate Student Fellowship | Mitra L Miri |
| Ford Foundation | Graduate Student Fellowship | Mitra L Miri |
| Human Frontiers | Postdoctoral Fellowship | Martin Vinck |
| Rubicon Fellowship | Postdoctoral Fellowship | Martin Vinck |
| National Eye Institute | EY022951 | Jessica A Cardin |
| Alfred P. Sloan Foundation | Fellowship | Jessica A Cardin |
| Swebilius Foundation | Grant | Jessica A Cardin |
| National Eye Institute | EY026878 | Jessica A Cardin |
| Esther A. and Joseph Klingenstein Fund | Fellowship | Jessica A Cardin |
| Whitehall Foundation | Grant | Jessica A Cardin |

The funders had no role in study design, data collection and interpretation, or the decision to submit the work for publication.

### Author contributions

Mitra L Miri, Formal analysis, Methodology, Writing—original draft, Writing—review and editing; Martin Vinck, Formal analysis, Writing—original draft, Writing—review and editing; Rima Pant, Data curation, Formal analysis; Jessica A Cardin, Conceptualization, Resources, Supervision, Funding acquisition, Methodology, Writing—original draft, Project administration, Writing—review and editing

### Author ORCIDs

Jessica A Cardin [iD] http://orcid.org/0000-0002-8209-5466

### Ethics

Animal experimentation: This study was performed in accordance with the recommendations in the Guide for the Care and Use of Laboratory Animals of the National Institutes of Health. All experiments were approved by the Institutional Animal Care and Use Committee of Yale University (#2015-11317).

### Decision letter and Author response

Decision letter https://doi.org/10.7554/eLife.40750.020
Author response https://doi.org/10.7554/eLife.40750.021

## Additional files

### Supplementary files
• Transparent reporting form
DOI: https://doi.org/10.7554/eLife.40750.018

## Data availability

Source data files are included for each figure and supplemental figure. Raw data are too large (750Gb) to include within the manuscript but are available on request as compiled sets of raw data and intermediate analysis files.

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
