## [Decision Letter]

[Editors’ note: a previous version of this study was rejected after peer review, but the authors submitted for reconsideration. The first decision letter after peer review is shown below.]

Thank you for choosing to send your work, "Altered hippocampal interneuron activity precedes ictal onset", for consideration at *eLife*. Your initial submission has been assessed by Gary Westbrook in consultation with a member of the Board of Reviewing Editors. Although the work is of interest, we regret to inform you that the findings at this stage are too preliminary for further consideration at *eLife*.

The two reviews are below for your consideration. In discussion with the reviewers and editor, we agreed that the work needs to be expanded to look at spontaneously occurring seizures to be of high impact, and further that the activity subsequent to the first "spike" is also of key relevance. This expansion would require major revisions, reworking the experimental design and analysis. We appreciate that the topic is of interest and the general approach could be useful.

*Reviewer #1:*

This report uses optogenetic tagging to examine, in two models of chemoconvulsant induced seizures, the general activity levels of two classes of hippocampal interneurons in the period leading up to seizures. The approach is solid, using careful methods to fully document the changes in PV, SST and non-PV/SST (presumed pyramidal cells). Results convincingly show that there is very little evidence for loss of inhibition in this critical preictal period. This is an area of controversy with some recent evidence suggesting failure of inhibition might play a role in ictogenesis, and this paper could have an important impact. The data, while technically convincing, are limited in some ways, that reduce overall impact. First, while two models are used, they are both acute chemoconvulsant induced seizures, and so might tell us little regarding the mechanisms leading up to spontaneously occurring seizures in epileptic subjects. Second, the results are obtained under anesthesia, and no discussion is provided regarding this important confound. Finally, the discussion is extremely terse, providing little insight regarding overall context of findings, potential limitations, or key future directions.

Essential revisions:

1) SFC. The PV SFC/frequency relationship is multipeaked, with preseizure changes not only in the 20-28Hz range, but also across the entire frequency band, with notable peak drop outs at 50, 80 and 100Hz.

2) The slope modulation of SST cells is potentially quite interesting and novel, but the relevance is not very well developed by the others. Similar for burstiness findings.

3) The four preictal periods are poorly defined. In particular, no statistics (range, median, etc.) seem to be provided for the overall total preictal periods, making it somewhat difficult to assess whether this is a reasonable cross-animal grouping.

4) Probably much more interesting things happen once the ictal activity starts to occur – this is arguably still a time in which the seizure is progressing, so presumably interneuron/principal cell activity continues to evolve.

5) The lack of difference in systematic spike shape between RS and PV cells is very surprising.

*Reviewer #2:*

The paper by Miri et al. explores the question of how PV and SST interneuron population function is affected during seizures. To this end the authors do a good job of developing two seizure models and examining how their firing is affected during preictal periods. The authors use state of the art methods to identify the interneuron populations combining single unit recordings with optogenetic stimulation, such that short latency responding cells can be distinguished by their responses. Importantly, although rather a sidebar to the central story the authors indicated that using spike width to classify neurons (particularly FS interneurons) is not particularly reliable. This secondary point aside, they do a very nice job of examining firing of PV, SST and RS (presumably largely pyramidal) neurons during preictal periods. Their major finding here is that PV but not SST interneurons alter their firing during preictal periods. They then go on to examine the inhibitory efficacy of PV and SST interneurons in preictal periods through optogenetic stimulation and examining their effect on pyramidal cells, which remains strong. From this they conclude that while interneuron function is retained during seizures, the PV population at least is decoupled from their normal function. In general, this is a very good paper that would make a good addition to *eLife* after some further experimentation is completed.

Essential revisions:

1) In Figure 1E I would like to see the change in firing rates for PV, SST, and RS after ictal onset. I realize that given the high level of activities recruited during seizures isolating single units may prove difficult to isolate. That said the advent of new methods where optogenetic methods are coupled with patch clamp methods make such an approach tenable and well within the skill set of this group. Even a limited set of patch recordings would greatly increase my enthusiasm for this work, especially as it would provide information as to the afferent IPSC/EPSCs within interneurons during preictal and ictal firing periods.

2) They tested the firing rates during the "preictal" period for two chemical models (PTZ and pilocarpine). I am a bit concerned that the progressive changes they see in the preictal period may be confounded by the kinetics of the arrival of the IP injection of the drugs to the brain rather than reflecting something that happens biologically before spontaneous seizures. I think it would add considerably to this study if they recorded the firing rate changes preceding a spontaneous seizure (1 week after the pilocarpine injection) to determine if the observed changes in firing rates are consistent between the two models. I concede that this experiment is technically challenging, and their system may not be stable enough to allow 1-week post-seizure induction survival. Nonetheless, demonstrating preictal periods even a few hours after drug treatment would help allay my concerns that their results are marred by injection artifact.

3) I think the paper would benefit from a more direct analysis of the decoupling of PV interneurons during the preictal period. As I suggested above, efforts to do some whole cell recording from PV and SST interneurons, identified using optogenetic stimulation would certainly increase the impact of their paper. At the very least, a further discussion of the implications of the uncoupling they observe in PV cells is needed. Since the hippocampus is tightly linked network and the PV cells are strongly connected to both the SST and RS populations, I was surprised that changes to the PV frequency spectrum of PV cells do not end up being reflected in the firing of other cells in the network. Given the previous work of the senior author demonstrating that PV cell activation can induce Γ band activity, further explanation why such a strong change in their firing rhythms are not reflected in the overall network activity warrants discussion. It would be helpful if the authors discuss this point and perhaps include a model as to how they think these changes lead to a seizure and network changes.

[Editors’ note: what now follows is the decision letter after the authors submitted for further consideration.]

Thank you for resubmitting your work entitled "Altered hippocampal interneuron activity precedes ictal onset" for further consideration at *eLife*. Your revised article has been favorably evaluated by Gary Westbrook as the Senior and Reviewing Editor and two reviewers.

Summary:

This revised manuscript investigates the differing roles of PV and SST interneurons in two models of acute pharmacologically induced seizures, using sophisticated and well thought out experimental methods. The study addresses an important area of controversy regarding the cellular mechanisms of ictogenesis, with some studies proposing failure of inhibition or depolarization block, and others implying that interneurons themselves are the trigger. The results show that inhibition remains intact in the preictal period, and that the two types of interneurons demonstrate reduced activity and sensitivity to input during ictal periods, along with cell-type specific differences in LFP coupling, suggesting that the two interneuron classes play different roles. The use of two chemoconvulsant models with different mechanisms of action is a strength. There is also a rigorous and much-needed demonstration of the pitfalls involved in distinguishing cell-type specific waveforms in extracellular recordings. This revised manuscript also now includes two pieces of highly novel data. First, the main result is validated in a limited number of recordings in non-anesthetized animals. Secondly, the authors provided very interesting results regarding continued evolution of neural activities past the preictal period. Overall, this is a convincing data set that will be of general interest. The paper provides important data regarding inhibitory activity in the time period leading up to ictal recruitment, and the observation of preictal/early ictal decoupling of PV firing from LFP β rhythms (20-28 Hz) is also an important new contribution.

Having said that, there remain some concerns regarding approach and interpretation.

1) We agree that injury induced, or genetic models would be much more difficult to study than acute chemoconvulsant seizures, especially given their variable and unpredictability. Nevertheless, the Abstract, Introduction, and Discussion section need to emphasize this point and must include the rationale included in the rebuttal. In particular, the seizure models must be listed in the Abstract e.g. by inserting the word chemoconvulsive before "seizure induction". There must be caution exercised in translating these results to spontaneous seizures occurring in the context of chronic epilepsy (i.e. in patients). This limitation should be mentioned in the Discussion section.

2) Despite the high quality of the results, some aspects of the paper are still underdeveloped. The last sentence of the introduction provides some observational results but does not put them into any particular context. The Results section ends on a similarly flat note.

3) The four preseizure periods could be better rationalized and/or explained in the results. It would help with clarity if it simply stated that the preseizure period was divided into four equally-spaced quartiles for each animal based on the latency between chemoconvulsant administration and seizure onset. What is a bit confusing is that the text indicates "equal duration" but this wording presumably does not apply across animals.

4) The definition of seizure onset is not clear, and it potentially confounds the results. To be fair to the authors, there is considerable controversy regarding the definition of an electrographic seizure, and how to determine when a specific site is recruited into an ongoing seizure. Nevertheless, the analysis is highly sensitive to these questions, and they should be taken into account in interpreting study results.

It is possible that the "ictal" period that is assessed may be composed in part or entirely of the discharging that is seen prior to ictal recruitment (i.e. penumbra). In Figure 1, the recruitment transition appears to take place well over a minute after ictal onset (based on this low resolution trace), so it would not have been included in the analysis which is limited to the first 60 seconds after onset. In Figure 2, the recruitment transition appears to occur during ictal period 4, in which there is a drop in PV and SST firing to baseline levels despite continued elevated excitatory firing. It would be very interesting to compare PV and SST activity to either side of this transitory event, if those data are available.

5) There is a mismatch between the time window of inhibitory failure vs. the time windows used for analysis of preictal and ictal periods. Inhibitory failure, as previously demonstrated in animal and human studies, is an abrupt event that occurs in < 2 seconds at the time of ictal recruitment. For this and reasons related to uncertainty of defining seizure onset, the conclusion in the Abstract and first paragraph of the Discussion section that "onset of ictal activity is not due to loss of firing … or failure of synaptic inhibition" is too strongly worded.

---

## [Author Response]

Reviewer #1:This report uses optogenetic tagging to examine, in two models of chemoconvulsant induced seizures, the general activity levels of two classes of hippocampal interneurons in the period leading up to seizures. The approach is solid, using careful methods to fully document the changes in PV, SST and non-PV/SST (presumed pyramidal cells). Results convincingly show that there is very little evidence for loss of inhibition in this critical preictal period. This is an area of controversy with some recent evidence suggesting failure of inhibition might play a role in ictogenesis, and this paper could have an important impact. The data, while technically convincing, are limited in some ways, that reduce overall impact. First, while two models are used, they are both acute chemoconvulsant induced seizures, and so might tell us little regarding the mechanisms leading up to spontaneously occurring seizures in epileptic subjects.

We agree with the reviewer that the selection of seizure models is a challenging issue. Chemoconvulsant models offer several advantages for detailed circuit examination in vivo. They are reliable and consistent across animals and have provided relevant insights into seizure pathology in previous work. One of our goals is to better understand the transition from healthy to pathological neural activity. Acute seizure induction allows us to separate the pattern of interneuron activity and inhibitory influence in the initial transition to ictal activity from alterations in interneuron activity that are known to occur as a result of seizure-induced cell death or sprouting.

In addition, although there are induced and genetic mouse models of seizure, each presents a different problem for data collection and interpretation. Because even clustered seizures in the pilocarpine model are unpredictable and spaced relatively far apart in time (e.g., Henderson et al., 2014), it is not possible to obtain sufficient ChR2-tagged interneuron recordings specifically at seizure onset to allow a rigorous analysis. In turn, most genetic models of epilepsy have known deficits in interneuron function (e.g., Tai et al., 2014; Adotevi and Leitch, 2016; Hedrich et al., 2014; Peñagarikano et al., 2011), and therefore are not suitable for understanding interneuron contributions to initial ictal transition in a healthy network. For these reasons, we feel that the models we use in the current work are both informative and well suited to the experimental goals.

Second, the results are obtained under anesthesia, and no discussion is provided regarding this important confound.

We thank the reviewer for providing the push to address this issue. We have now added recordings of identified interneurons and regular spiking cells in awake behaving animals during seizure initiation. We find increased PV and SST interneuron activity prior to ictal onset, with a sharp increase in PV firing immediately before the first ictal event. In addition, we find an altered temporal pattern of PV, but not SST, firing, with a preictal increase in the regularity of spiking.

These data extend our findings in two key ways. First, our findings in anesthetized and awake animals are very similar, suggesting that the anesthetized paradigm is a useful model. Second, these data add substantially to the novelty of our manuscript, as there are no previous studies of identified hippocampal interneurons during seizure in awake animals. We have also included a more complete discussion of the anesthetized paradigm.

Finally, the discussion is extremely terse, providing little insight regarding overall context of findings, potential limitations, or key future directions.

We have extensively revised the Discussion section and added details on context and relevance, limitations of the current model, and key next steps.

Essential revisions:1) SFC. The PV SFC/frequency relationship is multipeaked, with preseizure changes not only in the 20-28Hz range, but also across the entire frequency band, with notable peak drop outs at 50, 80 and 100Hz.

The reviewer is correct that there are multiple peaks in the SFC plots for PV interneurons. However, only the frequency band around the 20-28Hz range showed a significant drop, as highlighted by the additional analysis in Figure 4—figure supplement 1. We have revised the presentation of data in Figure 4 to better highlight changes in SFC between baseline conditions and the late preictal period.

2) The slope modulation of SST cells is potentially quite interesting and novel, but the relevance is not very well developed by the others. Similar for burstiness findings.

We have added more in-depth discussion of these findings and added context based on recently published work from other groups on SST control of pyramidal cell bursting in the hippocampus.

3) The four preictal periods are poorly defined. In particular, no statistics (range, median, etc.) seem to be provided for the overall total preictal periods, making it somewhat difficult to assess whether this is a reasonable cross-animal grouping.

The description of the preictal periods has been clarified in the text. In addition, we have included new plots (Figure 1—figure supplement 2) to clearly show the distribution of latencies to ictal onset. We have also added new plots to show that the cross-animal variation in the total duration of the preictal period does not affect the outcome of the analyses (Figure 1—figure supplement 2 and Figure 5—figure supplement 1).

We have further changed our statistical analysis strategy throughout the manuscript to be more rigorous. Using a Bonferroni correction, we have changed the threshold value for significance for the analyses comparing neural activity across the preictal periods to account for the use of repeated comparisons. This is also now better explained in the Materials and methods section and the legends.

4) Probably much more interesting things happen once the ictal activity starts to occur – this is arguably still a time in which the seizure is progressing, so presumably interneuron/principal cell activity continues to evolve.

We thank the reviewer for highlighting this aspect. Holding interneuron recordings after ictal onset is very challenging, but we were indeed able to follow a subset of interneurons after ictal onset (new Figure 2). We find that interneuron firing rates return to baseline levels, but regular spiking neurons continue to fire at elevated rates, suggesting that the change in interneuron activity is transient and specific to the transition from preictal to ictal activity. In comparison, the ictal period is associated with elevated excitatory, but not inhibitory, activity.

5) The lack of difference in systematic spike shape between RS and PV cells is very surprising.

We agree with the reviewer this result is novel. This comparison has not been made by previous studies, presumably because many previous datasets of identified hippocampal interneurons are quite small, with <10 cells per class. Previous work using waveform characteristics as the sole identifier of inhibitory interneuron identity has reported much more variable patterns of firing by putative FS interneurons at ictal onset than we find in the current work. To further support our claims, we have added new quantification of the laminar position of each recorded cell class in the hippocampus (Figure 1—figure supplement 1). We show that in the rostral portion of the hippocampus, where our recordings were made, nearly 100% of each interneuron class not only exclusively expresses one of the two genetic markers (PV or SST), but also has a cell body location consistent with the identified cell class. As we now suggest in the revised Discussion section, our data may explain some of this previous variability as the result of poorly identified or mixed cell classes.

Reviewer #2:[…]Essential revisions:1) In Figure 1E I would like to see the change in firing rates for PV, SST, and RS after ictal onset. I realize that given the high level of activities recruited during seizures isolating single units may prove difficult to isolate. That said the advent of new methods where optogenetic methods are coupled with patch clamp methods make such an approach tenable and well within the skill set of this group. Even a limited set of patch recordings would greatly increase my enthusiasm for this work, especially as it would provide information as to the afferent IPSC/EPSCs within interneurons during preictal and ictal firing periods.

To address this point, we have added the new Figure 2, which shows the change in firing rates for PV, SST, and RS cells after ictal onset. We find that interneuron firing rates return to baseline levels, but regular spiking neurons continue to fire at elevated rates, suggesting that the change in interneuron activity is transient and specific to the transition from preictal to ictal activity. In comparison, the ictal period is associated with elevated excitatory, but not inhibitory, activity. We agree with the reviewer that intracellular recordings would provide additional insights into synaptic input to interneurons during this period. However, in our extensive experience, this is a prohibitively difficult experimental approach to implement. Hippocampal interneurons are quite sparse, giving low yield for a single pipette, and changes in vascular dilation and tissue instability preclude holding high-quality patch recordings during ictal onset in vivo.

2) They tested the firing rates during the "preictal" period for two chemical models (PTZ and pilocarpine). I am a bit concerned that the progressive changes they see in the preictal period may be confounded by the kinetics of the arrival of the IP injection of the drugs to the brain rather than reflecting something that happens biologically before spontaneous seizures. I think it would add considerably to this study if they recorded the firing rate changes preceding a spontaneous seizure (1 week after the pilocarpine injection) to determine if the observed changes in firing rates are consistent between the two models. I concede that this experiment is technically challenging, and their system may not be stable enough to allow 1-week post-seizure induction survival. Nonetheless, demonstrating preictal periods even a few hours after drug treatment would help allay my concerns that their results are marred by injection artifact.

As suggested by the reviewer, the paradigms do not allow for post-seizure survival. In addition, because even clustered seizures in the pilocarpine model happen very infrequently (e.g., Henderson et al., 2014), it is not practically possible to obtain sufficient recordings from identified interneurons during seizure onset to perform a rigorous analysis. Previous work, now covered in the discussion, suggests that the kinetics of drug arrival in the brain do not match the trajectories of altered activity observed in the hippocampus. We have also now added new data from awake behaving animals showing the same results. In addition, the mechanisms of action of pilocarpine and PTZ are quite distinct and have different intrinsic kinetics (Huang et al., 2001; Turski et al., 1989; Ramzan and Levy, 1985; Yonekawa et al., 1980), whereas we find very similar neural trajectories in both cases. Finally, the drugs are presumably still present immediately after ictal onset, when we observe a rapid decline in interneuron activity and sustained firing of RS cells.

3) I think the paper would benefit from a more direct analysis of the decoupling of PV interneurons during the preictal period. As I suggested above, efforts to do some whole cell recording from PV and SST interneurons, identified using optogenetic stimulation would certainly increase the impact of their paper. At the very least, a further discussion of the implications of the uncoupling they observe in PV cells is needed. Since the hippocampus is tightly linked network and the PV cells are strongly connected to both the SST and RS populations, I was surprised that changes to the PV frequency spectrum of PV cells do not end up being reflected in the firing of other cells in the network. Given the previous work of the senior author demonstrating that PV cell activation can induce Γ band activity, further explanation why such a strong change in their firing rhythms are not reflected in the overall network activity warrants discussion. It would be helpful if the authors discuss this point and perhaps include a model as to how they think these changes lead to a seizure and network changes.

We regret that our original discussion of decoupling was not very clear. We have extensively rewritten the Results section and Discussion section to better synthesize our findings. The observation that PV cells increase their firing but become more regular and lose their coherence with the low gamma band is indeed quite interesting. In combination with the optogenetics findings, these results suggest that the overall level of interneuron influence is largely maintained during the preictal period, but the fine timing of inhibitory spikes is disrupted. This loss of PV timing may release RS cell spiking from tight temporal control over firing patterns. Over the same time period, the SST interneurons become less responsive to input. Because SST cells play a key role in restricting the bursting of excitatory pyramidal neurons, the loss of appropriately recruited SST activity may nonlinearly amplify excitatory activity in hippocampal circuits. Our results suggest a model in which the transition to ictal activity is associated with a disruption of the normal circuit operations of both PV and SST interneurons, potentially a two-pronged insult to the organization of hippocampal activity. We have further added a discussion of the potential compensatory role of the observed preictal increase in overall interneuron firing rates in transiently restricting the pathological recruitment of excitatory spiking.

[Editors’ note: what now follows is the decision letter after the authors submitted for further consideration.]

[…]Having said that, there remain some concerns regarding approach and interpretation.1) We agree that injury induced, or genetic models would be much more difficult to study than acute chemoconvulsant seizures, especially given their variable and unpredictability. Nevertheless, the Abstract, Introduction and Discussion section need to emphasize this point and must include the rationale included in the rebuttal. In particular, the seizure models must be listed in the Abstract e.g. by inserting the word chemoconvulsive before "seizure induction". There must be caution exercised in translating these results to spontaneous seizures occurring in the context of chronic epilepsy (i.e. in patients). This limitation should be mentioned in the Discussion section.

We agree with the reviewers that one should be cautious in linking the results from chemoconvulsive and spontaneous seizures. We have added ‘chemoconvulsant’ to the description in the Abstract. We note that these issues are rarely discussed in primary data papers, and we thank the reviewers for the opportunity to include a more complete description. We have further expanded the description of the rationale for choosing the chemoconvulsive models in the Introduction and have also added a cautionary note in the Discussion section:

Introduction:

“Here we used optogenetic tools to identify, track, and probe two distinct populations of hippocampal interneurons, the PV- and SST-expressing cells, in two models of acute chemoconvulsive seizure initiation in vivo. Chemoconvulsant models are reliable and consistent across animals and provide a window into the transition from healthy to pathological neural activity, separating interneuron activity and inhibitory influence in the initial transition to ictal activity from alterations in interneuron activity that are known to occur as a result of seizureinduced cell death or sprouting.”

Discussion section:

“We examined interneuron activity in two chemoconvulsant models of acute induction of status epilepticus. These models provide insight into the mechanisms by which normal, healthy neural circuits transition to pathological patterns of activation, although the precise commonality of circuit-level mechanisms underlying induced and spontaneous seizures has not been fully explored.”

2) Despite the high quality of the results, some aspects of the paper are still underdeveloped. The last sentence of the introduction provides some observational results but does not put them into any particular context. The Results section ends on a similarly flat note.

We have revised the end of the Introduction and the final statement of the Results section to add context and provide more specific interpretation.

3) The four preseizure periods could be better rationalized and/or explained in the results. It would help with clarity if it simply stated that the preseizure period was divided into four equally-spaced quartiles for each animal based on the latency between chemoconvulsant administration and seizure onset. What is a bit confusing is that the text indicates "equal duration" but this wording presumably does not apply across animals.

We have clarified the description in the Results section as suggested:

“In an initial series of experiments, we assessed the spontaneous activity of these three cell classes during a baseline period and four preictal periods leading into PTZ-induced seizure under light anesthesia (Figure 1D). For each animal, the preseizure period was divided into four equally spaced quartiles based on the latency from chemoconvulsant administration to ictal onset.”

4) The definition of seizure onset is not clear, and it potentially confounds the results. To be fair to the authors, there is considerable controversy regarding the definition of an electrographic seizure, and how to determine when a specific site is recruited into an ongoing seizure. Nevertheless, the analysis is highly sensitive to these questions, and they should be taken into account in interpreting study results.It is possible that the "ictal" period that is assessed may be composed in part or entirely of the discharging that is seen prior to ictal recruitment (i.e. penumbra). In Figure 1, the recruitment transition appears to take place well over a minute after ictal onset (based on this low resolution trace), so it would not have been included in the analysis which is limited to the first 60 seconds after onset. In Figure 2, the recruitment transition appears to occur during ictal period 4, in which there is a drop in PV and SST firing to baseline levels despite continued elevated excitatory firing. It would be very interesting to compare PV and SST activity to either side of this transitory event, if those data are available.

We agree with the reviewer that identifying ictal onset is a matter of continuing contention. Recent methods of seizure detection in rodents have included thresholding of the LFP or EEG, hand-tuning based on signal features, and measurement of power spectral coherence and entropy, each of which gives distinct results for ictal onset. In the current study, we used a quantitative measure of LFP activity, evaluated by z-scoring, to identify the earliest timepoint of pathological activity. This quantitative estimate largely coincided with the initial ictal spike. Our analysis thus focuses on the early phase of pathological activity, prior to full ictal recruitment. However, there are compelling reasons to look at this early period activity, as initiating events in the local circuit arise well prior to ictal recruitment. We were unfortunately not able to hold recordings of identified interneurons through ictal transition due to mechanical instability. In addition, there was some variability in the timing of recruitment transition across animals.

However, we fully recognize the limitations of this approach, and thus, in ongoing work we are examining the activity of identified interneurons during later ictal periods using 2-photon imaging, which does not provide precise spike times but is stable over longer periods of pathological activity. We have added text in the discussion to further highlight these issues and to emphasize that our current analysis focuses on relatively early events in the progression from normal activity to ictal recruitment:

Discussion section:

“Disruption of the excitatory-inhibitory balance profoundly alters circuit function (Fritschy, 2008; Ziburkus et al., 2013) and is hypothesized to play a significant role in the pathophysiological patterns of brain activity leading to the onset of seizure. The transition to ictal activity has been characterized in a variety of ways in rodents, including by manual validation (Henderson et al., 2014), threshold detection of increased EEG power (Khoshkhoo et al., 2017), and combined measurement of EEG power trajectory and spectral features (Krook-Magnuson et al., 2014). However, the patterns of activity underlying these transitions from stable to epileptiform activity are not well understood.”

In the current study, where we utilized an acute chemoconvulsant seizure induction, the onset of the ictal period was defined quantitatively by increased LFP amplitude, which largely corresponded to the first ictal spike. These early events occurred in advance of full ictal recruitment.”

5) There is a mismatch between the time window of inhibitory failure vs. the time windows used for analysis of preictal and ictal periods. Inhibitory failure, as previously demonstrated in animal and human studies, is an abrupt event that occurs in < 2 seconds at the time of ictal recruitment. For this and reasons related to uncertainty of defining seizure onset, the conclusion in the Abstract and first paragraph of the Discussion section that "onset of ictal activity is not due to loss of firing … or failure of synaptic inhibition" is too strongly worded.

We have softened the wording of the Abstract:

“Our findings suggest that the onset of ictal activity is not associated with loss of firing by these interneurons or a failure of synaptic inhibition but is instead linked with disruptions of the respective roles these interneurons play in the hippocampal circuit. “

We note that the Discussion section states that ‘the initial changes in network activity may not be associated with either an overt failure of inhibition or unconstrained excitation’, rather than stating that ‘ictal activity is not due to a failure of inhibition’. As noted in the response to point 4 above, the initial events that are the focus of the current manuscript occur earlier than many of the more electrographically prominent elements of seizure, such as broad ictal recruitment. Our data suggest that these early initiating events are not associated with a failure of inhibition, but do not allow us to draw conclusions about inhibitory function later in the ictal period. We have added this point to the Discussion section and emphasized the early timepoint:

“Overall, these results suggest that the initial changes in network activity leading to early stages of seizure initiation may not be associated either with an overall failure of GABAergic inhibition from PV or SST cells or with unconstrained excitation. However, we note that our findings do not preclude additional impairments in inhibitory function at the time of ictal recruitment or later in the ictal period.”